# CT-Based Risk Stratification for Intensive Care Need and Survival in COVID-19 Patients—A Simple Solution

**DOI:** 10.3390/diagnostics11091616

**Published:** 2021-09-04

**Authors:** Clarissa Hosse, Laura Büttner, Florian Nima Fleckenstein, Christina Maria Hamper, Martin Jonczyk, Oriane Scholz, Annette Aigner, Georg Böning

**Affiliations:** 1Institute of Radiology, Charité-Universitätsmedizin Berlin, Corporate Member of Freie Universität Berlin and Humboldt-Universität zu Berlin, Charitéplatz 1, 10117 Berlin, Germany; clarissa.hosse@charite.de (C.H.); florian.fleckenstein@charite.de (F.N.F.); christina-maria.hamper@charite.de (C.M.H.); martin.jonczyk@charite.de (M.J.); oriane.scholz@charite.de (O.S.); georg.boening@charite.de (G.B.); 2Berlin Institute of Health at Charité-Universitätsmedizin Berlin, Charitéplatz 1, 10117 Berlin, Germany; annette.aigner@charite.de; 3Institute of Biometry and Clinical Epidemiology, Charité-Universitätsmedizin Berlin, Corporate Member of Freie Universität Berlin and Humboldt-Universität zu Berlin, 10117 Berlin, Germany

**Keywords:** COVID-19, SARS-CoV-2, CT, quantification, risk assessment, intensive care, resource allocation, developing countries

## Abstract

We evaluated a simple semi-quantitative (SSQ) method for determining pulmonary involvement in computed tomography (CT) scans of COVID-19 patients. The extent of lung involvement in the first available CT was assessed with the SSQ method and subjectively. We identified risk factors for the need of invasive ventilation, intensive care unit (ICU) admission and for time to death after infection. Additionally, the diagnostic performance of both methods was evaluated. With the SSQ method, a 10% increase in the affected lung area was found to significantly increase the risk for need of ICU treatment with an odds ratio (OR) of 1.68 and for invasive ventilation with an OR of 1.35. Male sex, age, and pre-existing chronic lung disease were also associated with higher risks. A larger affected lung area was associated with a higher instantaneous risk of dying (hazard ratio (HR) of 1.11) independently of other risk factors. SSQ measurement was slightly superior to the subjective approach with an AUC of 73.5% for need of ICU treatment and 72.7% for invasive ventilation. SSQ assessment of the affected lung in the first available CT scans of COVID-19 patients may support early identification of those with higher risks for need of ICU treatment, invasive ventilation, or death.

## 1. Introduction

A new form of coronavirus was identified in Wuhan, China, in 2019 and named severe acute respiratory syndrome coronavirus 2 (SARS-CoV-2) [1,2]. The virus causes coronavirus disease 2019 (COVID-19), which rapidly became a pandemic. In August 2021, there were over 200 million confirmed cases of COVID-19 and more than four billion deaths worldwide [3]. COVID-19 has become a burden to health care systems all over the world and is still associated with significant mortality. The main transmission route of the virus is via droplet infection [4]. Infected patients may present with symptoms ranging from mild to severe pneumonia with acute respiratory distress syndrome (ARDS), lung failure, and death. Most patients with a severe clinical course of COVID-19 have comorbidities and risk factors, such as older age, smoking, hypertension, cardiovascular disease, and diabetes mellitus [5]. Fourteen percent of hospitalized patients require admission to an intensive care unit (ICU), and up to 17% of patients require mechanical ventilation [6,7]. Acute respiratory distress syndrome (ARDS) occurs in 60–70% of COVID-19 patients admitted to an ICU [8]. In the course of the pandemic, medical resources have become scarce, particularly in emerging and developing countries. Therefore, limited medical equipment and supplies such as ventilators and oxygen as well as medical staff must be used as efficiently as possible in order to avoid a collapse of the health care systems.

A multidisciplinary panel of pulmonologists and radiologist from 10 countries showed that computed tomography (CT) plays a crucial role in the diagnosis of COVID-19 pneumonia (67–100% sensitivity) [9]. Pulmonary involvement in COVID-19 is predominantly bilateral with peripheral distribution in the middle and lower lung regions with typical patterns including ground-glass opacities (GGO), consolidations, and crazy-paving [5,10,11]. Previous studies have already shown that findings on chest CT may help predict the course of the disease [12]. Recent studies using artificial intelligence (AI) and volumetry suggest that quantification of pulmonary involvement can predict a patient’s disease course, prognosis, and need for ICU care [11,13]. However, these methods are limited, as they require tools such as volumetry software or machine learning algorithms, which are not available all over the world, and comparison of different methods is difficult [14]. Furthermore, the use of such tools in general requires a number of additional clinical parameters as well as trained personnel [15]. Therefore, a simple method would be highly desirable, especially in countries where the medical infrastructure does not meet the standards of industrialized nations. A recent study presents a simple semi-quantitative (SSQ) method that outperforms subjective evaluation and might be used more widely than quantitative methods such as volumetry or AI [16].

The aim of the current study was to investigate the usability of the SSQ method in a larger study population during the second wave of COVID-19 in Germany, as it may help to optimize allocation of limited ICU and ventilation capacities in global health care systems in the next waves.

The principal conclusions are as follows:With the SSQ method, a 10% increase in affected lung area increased the risk for need of ICU treatment with an OR of 1.68 (95% CI: 1.41–2.03) and for invasive ventilation with an OR of 1.58 (95% CI: 1.35–1.87).A larger total affected lung area increased the cause-specific instantaneous risk of dying (HR 1.11 (95% CI: 1.01–1.23) independently of other risk factors.SSQ assessment of lung involvement in the first available CT scans of COVID-19 patients may support early identification of those in need for ICU treatment and invasive ventilation, yet well-trained radiologists achieve comparable results.

## 2. Materials and Methods

### 2.1. Study Design and Patient Population

This single-center study in a level-1 center was approved by the local ethics committee. Written informed consent was waived for this retrospective study. All patients included tested positive for SARS-CoV-2 (RT-PCR of nasopharyngeal and oropharyngeal swab samples) and underwent chest computed tomography (CT) during the second wave of the COVID-19 epidemic in Germany between August and December 2020. Exclusion criteria were a negative COVID-19 test, no CT examination, and patients with acute disease who had already been intubated (see Figure 1).

We documented initial symptoms, clinical course, comorbidities, date of death, indication for CT examination, and need of ICU and of invasive ventilation. The endpoints of the study were need for ICU admission, need for invasive ventilation, and death. Invasive ventilation included intubation, tracheostomy, and extracorporeal membrane oxygenation (ECMO) treatment. The indications for intensive care and invasive ventilation were based on clinical parameters (e.g., oxygen saturation, respiratory rate, blood pressure) in concordance with current guidelines, which consider admission to ICU of COVID-19 patients to be appropriate if SpO_2_ is below 90% and the patient has dyspnea, a respiratory rate >25–30/min, and systolic blood pressure <100 mmHg or elevated lactate levels [17]. In patients with hypoxemia or respiratory insufficiency, the first therapeutic option is the administration of oxygen via nasal tube, Venturi mask, and high-flow oxygen therapy [17]. In case of progressive deterioration of gas exchange and increased oxygen demand, continuous positive airway pressure (CPAP) therapy or noninvasive or invasive ventilation should be contemplated [17]. Some patients with a clinical indication were not put on invasive ventilation or were not admitted to the ICU due to existing living wills or because consent was refused by patients or relatives. In order to rule out possible bias resulting from such decisions, we chose the medical indication to determine the endpoint “need for invasive ventilation” and “need for ICU treatment”, regardless of whether or not the treatment was actually performed.

### 2.2. Subjective and SSQ Assessment of CT Scans

Lung involvement on initial chest CT scans was assessed by estimating the extent of consolidations and ground-glass opacities both subjectively and with the SSQ method. Furthermore, additional pathological findings such as lymphadenopathy, pleural effusions, as well as the distribution and accentuation patterns of lung infiltrates were noted using structured reporting. Subjectively, the extent of ground-glass opacities and consolidations was classified as follows: none, minor (0–33% lung area), moderate (34–66% lung area), or major (67–100% lung area). SSQ analysis was performed as described before [16]. Briefly, the area of affected lung parenchyma was measured in polygonal regions of interest (ROIs) in one image at three defined anatomical levels: aortic arch, tracheal bifurcation, and inferior end of xiphoid. For each level, the amounts of consolidations and ground-glass opacities were added and divided by the sum of total lung area in this region. Additionally, mean values of affected and total lung area across the three regions were calculated per case (see Figure 2).

### 2.3. CT Scan Parameters

CT examinations were performed using our clinical standard protocols on different CT scanners of our department. If COVID-19 was the only indication for the CT scan, the technical parameters of low-dose protocols (effective dose <1 mSv) were as follows [12]: Canon Aquilion Prime (Canon, Tokyo, Japan): 100 kVp voltage, 10–120 mA tube current, recon IR level AIDR 3D standard, recon section interval of 1 mm; GE Light-Speed VCT (GE Healthcare, Boston, MA, USA): 100 kVp voltage, 20–120 mA tube current, recon IR level ASIR 50, recon section interval of 0.625 mm. Accordingly, most CT scans were acquired during a single breath-hold without contrast agent administration. For patients with additional clinical questions to be answered by the CT scan (e.g., pulmonary embolism), the appropriate standard contrast-enhanced protocols were used. ROIs were defined at 5 mm slice thickness using a lung kernel.

### 2.4. Statistical Evaluation

The required sample size of 205 was calculated based on the AUC estimate for the SSQ method regarding the endpoint of ICU treatment as derived in a previous pilot study (85.6%) [16], such that the width of the 95% confidence interval (CI) is 85.6% ± 5%.

Clinical endpoints and other categorical variables are reported as absolute and relative frequencies, continuous variables as medians along with interquartile ranges. Risk factors for the need of invasive ventilation and ICU admission are analyzed using logistic regression models, where results are presented as odds ratios (ORs) along with 95% confidence intervals. Additionally, the diagnostic performance of the SSQ method is analyzed with receiver operating characteristic (ROC) curves and the respective area under the ROC curves (AUC). For time to death, a Cox proportional hazards regression was performed, where patients were censored at discharge. Statistical analysis was performed using R (R Core Team) including additional packages for data handling, plotting, and analysis [18].

## 3. Results

### 3.1. Epidemiologic Patient Data

Our study population included 265 patients. The median interval between onset of symptoms and a positive SARS-CoV-2 PCR test was 3 days (IQR 1–6 days). The median time between a positive test and CT examination was 0 days (IQR 0–4 days). CT examinations were most commonly performed to assess pulmonary involvement (*n* = 108, 41.4%) or dyspnea (*n* = 89, 34.1%) and to search for an inflammatory focus (*n* = 61, 23.4%). The mean duration of hospitalization was 9 days (IQR 11 days). Predominant symptoms were fever (64.8%), weakness (64.5%), and cough (51.3%). With regard to the frequency of symptoms, there were no significant differences between patients with/without the need for invasive ventilation or ICU admission.

More than half of patients did not need ICU treatment (*n* = 151, 57.0%). Most patients only needed nasal oxygen (*n* = 112, 42.7%), no ventilation at all (*n* = 56, 21.4%), or noninvasive ventilation, e.g., high-flow or CPAP therapy (*n* = 49, 18.7%). Invasive ventilation was used in 34 patients (13.0%), and 11 patients even required ECMO treatment (4.2%). A total of 137 patients had an indication for ICU treatment (51.5%), while 114 patients were actually admitted to the ICU (42.9%). Eighty-one patients had an indication for invasive ventilation (30.5%), but only 45 were intubated (16.9%).

The study patients had a median age of 68.0 years, and patients requiring ICU treatment (median 75.0 years) and invasive ventilation (median 70.0 years) were slightly older. The majority of patients were male (*n* = 163, 61.5%). Patients with a need for ICU treatment or invasive ventilation more often had comorbidities such as diabetes mellitus, hypertension, or chronic lung disease (Table 1).

### 3.2. Subjective CT Assessment

In the majority of chest CT examinations performed in our study population, the extent of pulmonary involvement by area was subjectively classified as moderate (40.0%) or minor (34.7%). Most patients needing ICU treatment and invasive ventilation later on were classified as moderate (ICU: 43.1%, invasive ventilation: 39.5%) or major (ICU: 27.7%, Invasive ventilation: 33.3%). Predominant findings in CT scans of all patients were ground-glass opacities (GGOs, 89.1%) and consolidations (74.3%). Just under one-quarter of the patients did not have any consolidations. In patients who needed to be intubated or transferred to the ICU in the further course, consolidations tended to be more extensive and were more often classified as major (invasive ventilation: 13.6% vs. 5.4% of patients; ICU admission: 13.1% vs. 2.3% of patients) (see Table 2). The same tendency was seen for GGOs: extensive GGOs were found in patients with a need for invasive ventilation or for ICU treatment. Other findings, such as pleural effusions or lymphadenopathy, were rare. The majority of patients had consolidations and GGOs predominantly involving basal (43.4%) or posterolateral (32.1%) pulmonary areas. There was a trend toward involvement of the posterolateral lung segments in patients who later needed invasive ventilation or ICU treatment. The most common distribution patterns were peribronchial (35.1%) and peripheral (38.9%) (see Table 2).

### 3.3. SSQ Assessment of CT Scans

Using the SSQ method, we found a median total affected lung area of 15.3%. Patients who later needed intensive care usually had the involvement of a larger total lung area (ICU admission: 22.6% vs. 8.2% lung area; invasive ventilation: 25.5 vs. 11.7% lung area) (see Table 3). In the total study population, the areal extent of the affected lung was largest at the level of the xiphoid (18.8% affected lung area). In patients who had to be intubated or transferred to the ICU during the course of their treatment, the median affected lung area was more than twice as large at all three levels. Of note, there was also a more extensive involvement of the cranial pulmonary segments in these patients (at the level of the aortic arch: ICU admission: 19.6% vs. 2.7% affected lung area; invasive ventilation: 19.6% vs. 2.7% affected lung area) (see Table 3).

### 3.4. Diagnostic Performance

In ROC analysis of the diagnostic performance of subjective assessment of CT scans, the AUC for the endpoint need for ICU admission was 70.1% (95% CI: 64.3–75.9%) versus 67.9% (95% CI: 61.3–74.6%) for need for invasive ventilation. For total affected lung area estimated with the SSQ method, the AUC for the endpoint need for ICU admission was 73.5% (95% CI: 67.6%–79.5%). Based on Youden’s index, an involvement of 14.3% of the total lung parenchyma was found to be the optimal cutoff, resulting in 65.6% specificity and 72.1% sensitivity. For the second endpoint, the need for invasive ventilation, the AUC of the SSQ method was 72.7% (95% CI: 66.3–79.1%). The cutoff identified with Youden’s index was 8.3% (resulting in 43.2% specificity and 88.9% sensitivity) (see Figure 3).

### 3.5. Risk Stratification

Logistic regression revealed that a 10% increase in the affected lung area estimated with the SSQ method was associated with higher odds of needing ICU treatment (OR 1.68 (95% CI: 1.41–2.03)). There was also an effect on the odds of needing invasive ventilation (OR 1.58 (95% CI: 1.35–1.87)). We found that male patients had higher odds of needing invasive ventilation (OR 2.25 (95% CI: 1.16–4.52)) or ICU treatment (OR 1.20 (95% CI: 0.68–2.15)). A pre-existing chronic lung disease also increased the odds for the need of invasive ventilation (OR 2.73 (95% CI: 1.32–5.73) or ICU treatment (OR 1.78 (95% CI: 0.90–3.60)). Smoking had a protective effect for the need of ICU admission (OR 0.54 (95% CI: 0.25–1.13) and for the need of invasive ventilation (OR 0.55 (95% CI: 0.23–1.24) (see Figure 4).

### 3.6. Survival Analysis

Seventy-two patients (27.2%) died during the study period, most of whom were male (mortality rate male/female: 29.4% vs. 23.5%). Patients who died were older (median age: 79 years; IQR 19.0 years vs. median age: 64.4 years; IQR 22.2 years) and more often had a history of chronic lung conditions such as fibrosis or emphysema (28.2% vs. 18.7% of patients).

The results of the Cox regression model showed that a higher percentage of involved total lung estimated with the SSQ method increased the instantaneous risk of death with HR 1.11 (95% CI: 1.01)). We also found that pre-existing chronic lung disease (HR 1.90 (95% CI: 1.12–3.22), age (HR 1.91 (95% CI: 1.53–2.38)), and male sex (HR 2.0 (95% CI: 1.17–3.44)) each increased the instantaneous risk of death independently of other potential risk factors. Smoking had a protective effect for death (HR 0.44 (95% CI: 0.21–0.94) (see also Figure 5).

## 4. Discussion

CT is an important tool for assessing the risk of patients with COVID-19. Associations between the extent of pulmonary involvement in CT and the severity of disease have been demonstrated using volumetric tools and approaches including AI [11,19,20,21]. However, these approaches have limitations, e.g., in terms of software availability and costs, which limit their widespread use, especially in structurally weaker regions [11]. Therefore, a simpler method that can be used everywhere without additional costs could make an important contribution to an efficient allocation of limited resources such as oxygen, ventilators, and other equipment during future waves of the pandemic.

Therefore, the aim of this study was to prove the principle of a ubiquitously available, simple semi-quantitative (SSQ) method for estimating the extent of pulmonary involvement in CT scans, which was first described during the first wave of COVID-19 infections in Germany [16], in a larger patient population.

### 4.1. Epidemiologic Patient Data

In our study population, 51.5% of patients needed intensive care and 30.5% needed invasive ventilation. Those rates are higher than reported in the literature for hospitalized patients with 14% requiring ICU admission, up to 17% needing mechanical ventilation, and around 7% requiring invasive ventilation [6]. This difference may be attributable to the fact that our hospital is a level-1-center, and many critically ill patients are transferred from other hospitals, in particular due to our expertise with ECMO treatment. The most commonly described symptoms of COVID-19 infections such as cough, fever, and dyspnea were confirmed in our patient population [6]. In our analysis, the majority of patients were male (61.5%), which is in line with previous studies suggesting that men affected with COVID-19 seem to have a more severe clinical course than women [22]. In our analysis, male patients had a higher risk of needing invasive ventilation (OR 2.25) or ICU treatment (OR 1.22). Furthermore, most of the deceased patients in our study were male (mortality rate male/female: 29.4% vs. 23.5%). It is noteworthy that testosterone is a co-regulator of ACE2 enzyme and may facilitate SARS-CoV-2 internalization [23]. Furthermore, low blood concentrations of testosterone, e.g., in older men, can cause endothelial dysfunctions as well as thrombosis and inflammation [24]. These factors may contribute to more severe courses of disease in general and thrombotic complications in particular.

### 4.2. Subjective CT Assessment

The typical CT imaging features of COVID-19 pneumonia such as bilateral, peripherally accentuated consolidations, and ground-glass opacities (GGOs) were again confirmed in our patient population [25]. In our study, GGOs and consolidations were more frequent in patients with a need for ICU admission and invasive ventilation, again confirming findings described in the literature [19]. Pleural effusions and lymphadenopathy were rare, both in our patients and in an earlier report [25].

### 4.3. Diagnostic Performance

Confirming preliminary results, we found the SSQ method for determining the extent of pulmonary involvement in CT to be slightly superior to subjective assessment in our analysis regarding the prediction of a patient’s COVID-19 course [16]. However, compared to data from the first wave of the epidemic in Germany, the advantage of the SSQ method shrunk (15). In the previous study, the diagnostic performance found for subjective assessment was 64.2% (95% CI: 45–83.3%) for the need for ICU admission and 64.6% (95% CI: 44.6–84.6%) for the need for invasive ventilation; the corresponding AUCs were 70.1% (95% CI: 64.3–75.9%) and 67.9% (95% CI: 61.3–74.6%), respectively. The observed improvement in subjective assessment is probably attributable to radiologists’ increasing experience with the interpretation of imaging findings in COVID-19 patients and the use of structured reporting. Various studies show that structured reporting can help to install a baseline in reporting and thus enhance diagnostic performance (28). Therefore, it is reasonable to assume that well-trained radiologists, working in a level-1 center with high case numbers of COVID-19 patients, can achieve a performance comparable to that of the SSQ method. For radiologists in hospitals and private practices seeing fewer COVID-19 patients, the SSQ method may still offer a good option to predict the need for invasive ventilation and ICU admission in COVID-19 cases.

### 4.4. Risk Stratification

We found that the extent of involved lung area increases the risk of needing ICU admission more markedly than the risk of needing invasive ventilation. This tendency is in line with a prior analysis showing that a 10% increase in the affected lung parenchyma area increased the instantaneous risk of invasive ventilation (hazard ratio (HR) = 2.00) and the instantaneous risk of ICU admission (HR = 1.73) (15). A similar conclusion, namely that quantification of lung opacification in COVID-19 measured in CT by deep-learning-based tools or volumetry can predict severity, was drawn by earlier investigators (6, 8, 27). As deep learning and volumetry are not available everywhere, a simple method is preferable, and good results were obtained using SSQ measurement in a previous pilot study (15).

In the present analysis, pre-existing chronic lung disease, older age, and male sex increased the risk of needing intensive care. Other studies also found that male patients suffered from more severe symptoms and showed higher mortality [26]. In addition to gender, age also significantly influences outcome in COVID-19 patients. Older patient age is also considered a risk factor for a severe course of COVID-19 infection [27]. In our population, the patients in need of invasive ventilation were nearly 10 years older than patients not requiring invasive ventilation. The same tendency was shown in patients in need of ICU admission. Castelnuovo et al. 2021 have shown that in addition to advanced age, conditions such as hypertension, underlying cardiovascular, pulmonary, or renal disease strongly influence patient outcome [28]. Our data also confirm the findings of a large-scale study in 10,021 hospitalized patients showing that pre-existing chronic pulmonary diseases are more frequent in patients needing invasive ventilation [6]. In a cohort such as in our level-1-center with older and sicker patients, there are more patients requiring invasive ventilation or ICU stay than in a small peripheral hospital. Furthermore, therapies may also have an impact on outcome in age groups. For example, studies have shown that the combination of darunavir/cobicistat was associated with higher mortality in the elderly [28], whereas heparin may have a productive effect [29]. Older patient age is also considered a risk factor for a severe course of COVID-19 infection [27]. In our population, the patients who needed invasive ventilation were nearly 10 years older than patients not requiring invasive ventilation. The same tendency was shown in patients in need of ICU admission. Our data also confirm the findings of a large-scale study in 10,021 hospitalized patients, showing that pre-existing chronic pulmonary diseases are more frequent in patients needing invasive ventilation [6].

SARS-CoV-2 not only causes pulmonary damage but also leads to cardiovascular complications including myocardial injury, myocarditis and pericarditis, arrythmia and cardiac arrest, cardiomyopathy, heart failure, cardiogenic shock, and coagulation abnormalities [30]. Our results also revealed that patients with hypertension were at higher risk for the need of ICU treatment (66.4% vs. 51.2%) and that patients with coronary artery disease (CAD) were at higher risk for the need of invasive ventilation (53.1% vs. 26.6%). These results are in line with previous reports that COVID-19 patients suffering from cardiovascular diseases have higher morbidity and mortality rates [31]. Different mechanisms have been postulated for cardiovascular involvement such as cytokine storm, activation of receptor angiotensin-converting enzyme 2 (ACE2), impact of antiviral drugs, hypoxemia, and stress [30,32].

Lung ultrasound as an alternative examination method has emerged over the last 20 years as a noninvasive and easily available tool for the rapid differential diagnosis of pulmonary diseases such as community-acquired pneumonia [33,34,35]. Lung ultrasound is particularly useful for monitoring the progression of COVID-19 disease, since it is possible with small, mobile devices that are easy to disinfect [36]. A limitation of ultrasound is that it is examiner-dependent and should preferably always be performed by the same examiner [36].

### 4.5. Survival Analysis

During the observation period, 27.2% of patients in our study population died. This is comparable to reported mortality rates in the literature, among them a large study conducted in Germany that found a mortality rate of 22% [6]. The same study also found a higher mortality rate in older compared to younger patients. Meta-analyses found that older age was associated with higher mortality [6]. We also found that older patients had an increased risk of death. Data from the literature also show that male patients are more likely to become severely ill and die twice as often as women [26].

In our study, a 10% increase in total affected lung area increased the instantaneous risk of dying with HR 1.11 (95% CI: 1.01–1.23). A study of 572 hospitalized patients showed the same trend; 70% of patients who had more than 50% total lung involvement were treated in the ICU or died within seven days while the rate was significantly lower in patients with less extensive lung involvement [37]. Our findings also suggest that chronic lung conditions are associated with a nearly two times higher instantaneous risk of death (HR 1.90 (95% CI: 1.12–3.22)).

### 4.6. Limitations

The main limitation of our study is its retrospective design. As a level-1 care center, we treat many critically ill patients referred from smaller hospitals. Therefore, our patient population is not representative of the general population. This may also explain the fact that smoking appeared to have a protective effect in our patient population. The same association is reported in some published studies, while others suggest that smoking is associated with poorer outcomes [16,38]. The second limitation is that the sample size calculations were based on previous pilot study results regarding the endpoints of invasive ventilation and transfer to the intensive care unit. However, as there was a high proportion of patients refusing intensive care and invasive ventilation in our population, the medical indications had to be chosen as endpoints in order to prevent a possible bias due to patients’ decisions. While exact time points are available for treatments actually performed, this is not the case for establishing indications; therefore, no time-to-event analysis could be performed with the two primary endpoints. A third limitation is that ROIs were placed manually, which is therefore to some extent subjective. Yet we used standardized levels of measurement and a predefined slice thickness of CT scans to ensure comparability between patients and also with the earlier pilot study [16]. As the appearance of COVID-19-related pulmonary changes depends on when a CT examination is performed, the fourth limitation of our study is the heterogeneity of CT examination time points with a median of 0 days between a positive test and the CT examination. Furthermore, 10.6% of symptomatic COVID-19 patients have been reported to have normal chest CT findings [39]. Normal CT scans are most common during the first five days of the disease (13.9–33.3% of patients) [40]. At later stages, normal chest CT findings become very rare (1.2–4% of patients) [40,41,42]. Recent studies suggest that nearly half of asymptomatic patients (46%) have a normal chest CT [43]. The subjective assessment of pulmonary involvement in the CT images is limited by the fact that it was performed as part of clinical routine by a resident and a specialist in consensus. All in all, this subjective method should contribute to highlighting the effectiveness of SSQ rather than being a stand-alone method. This limit should be further emphasized considering that it was used as a comparison to underline the importance and effectiveness of the SSQ method. Potential errors of subjective assessment, particularly in less trained/standardized clinics, underscore the importance of the SSQ method.

## 5. Conclusions

The SSQ method is a good tool for predicting the need for both invasive ventilation and intensive care therapy. Nevertheless, subjective assessment by well-trained radiologists can provide comparable performance. An increase in the affected lung area was associated with significantly higher odds for need of ICU treatment and invasive ventilation. SSQ-derived involvement of total lung area was associated with mortality independent of other risk factors. Therefore, the SSQ method could help improve efficient allocation of treatment especially in countries with limited health care resources and in institutions where routine in the interpretation of CT scans from COVID-19 patients is limited. Furthermore, it would be possible to predict the course of the disease at an early stage in order to provide critical patients with appropriate therapy as early as possible. The method could help to organize and allocate the capacities of intensive care beds, ventilators and staff. As an advantage, the method is easy to integrate into daily clinical practice; especially in developing countries where fully automated software solutions, such as AI-based approaches, for lung involvement analysis are not widely available.

## Figures and Tables

**Figure 1 diagnostics-11-01616-f001:**
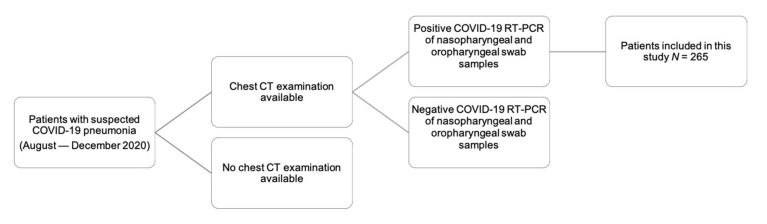
Patient population. We retrospectively enrolled 265 patients admitted to our central European level-1 university center. All patients included underwent a chest CT examination and tested positive for SARS-CoV-2 (RT-PCR of nasopharyngeal and oropharyngeal swab samples); patients with a negative swab test were excluded.

**Figure 2 diagnostics-11-01616-f002:**
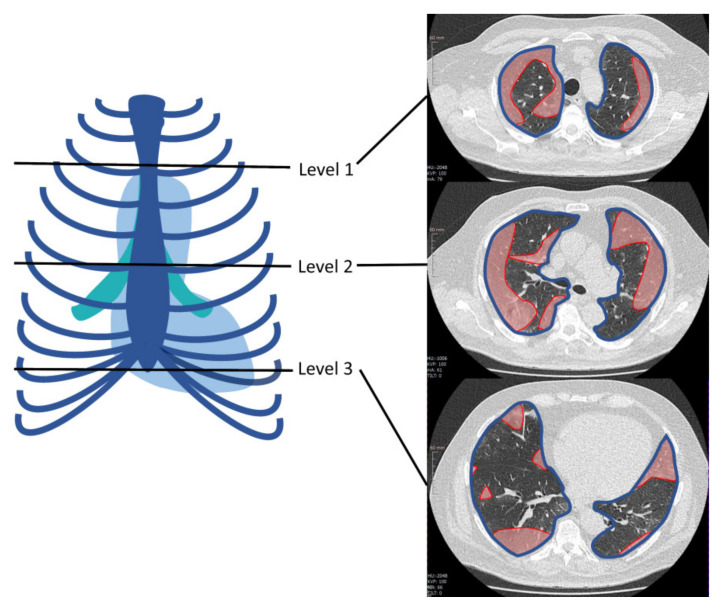
Example of region of interest (ROI)-based simple semi-quantitative (SSQ) determination of lung area involved in chest CT scans of COVID-19 patients at three predefined levels: aortic arch, tracheal bifurcation, and inferior end of xiphoid. In the axial image from each level, red indicates the area of involved lung parenchyma while the blue line outlines the total lung area at that level. Reprinted from Büttner et al. [16] with kind permission of MDPI.

**Figure 3 diagnostics-11-01616-f003:**
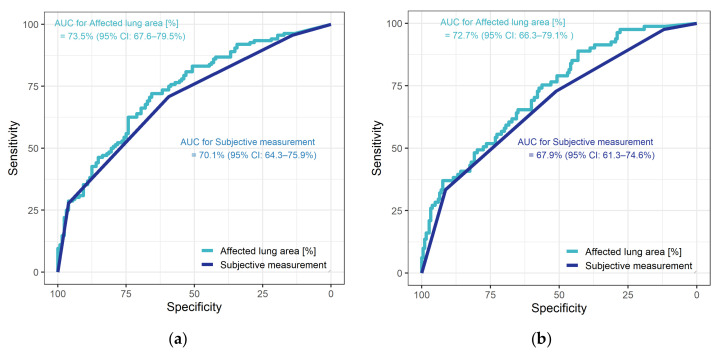
Diagnostic performance of subjective assessment and of the SSQ method for the two endpoints: (**a**) need for ICU treatment, (**b**) need for invasive ventilation.

**Figure 4 diagnostics-11-01616-f004:**
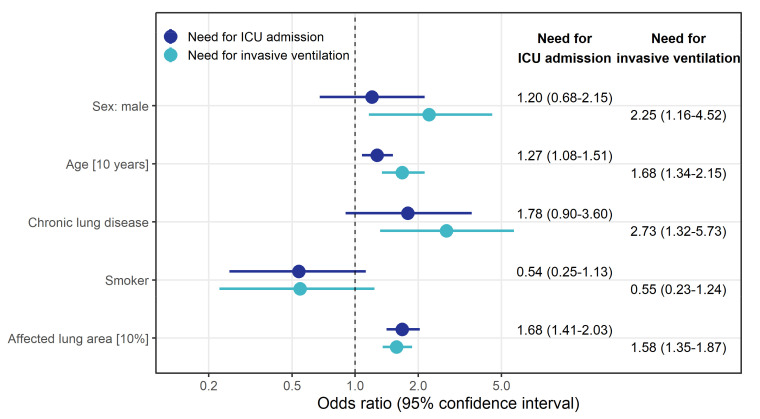
Odds ratio estimates along with 95% confidence intervals (CIs) derived from logistic regression model to identify factors associated with the risk for ICU admission (left column) and invasive ventilation (right column). The model included all variables displayed.

**Figure 5 diagnostics-11-01616-f005:**
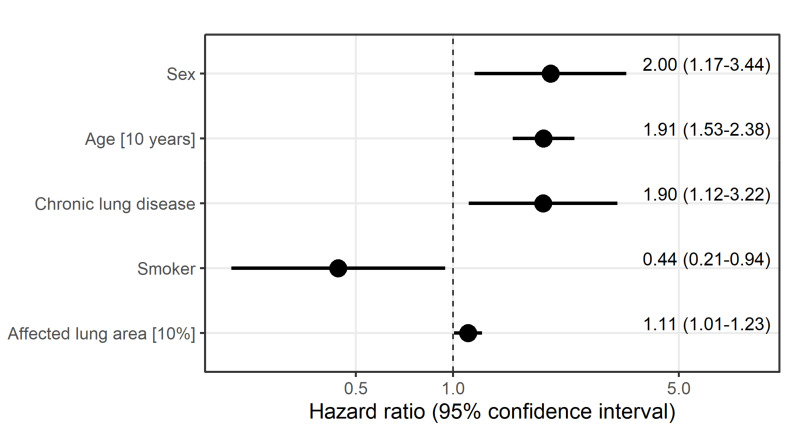
Hazard ratio estimates along with 95% confidence intervals derived from Cox regression model to identify factors associated with the instantaneous risk of death. All variables displayed were included in the model.

**Table 1 diagnostics-11-01616-t001:** Summary of demographic and other patient data by endpoint and in the total study population. Median patient age was 68.0 years, patients with need for ICU treatment (median 75.0 years) and need for invasive ventilation (median 70.0 years) were slightly older. Most patients were male (*n* = 163, 61.5%). Patients with need for ICU treatment or need for invasive ventilation more often had diabetes mellitus, hypertension, or a chronic lung disease. The most frequent symptoms were weakness (64.5%), fever (64.8%), and cough (51.3%). The majority of patients required nasal oxygen in the course of the disease (42.7%). Abbreviations: IQR: interquartile range, CAD: coronary artery disease, ECMO: extracorporeal membrane oxygenation.

	No Need for Invasive Ventilation (*n* = 184)	Need for Invasive Ventilation (*n* = 81)	No Need for ICU Treatment (*n* = 128)	Need for ICU Treatment (*n* = 137)	Total (*n* = 265)
Sex and Age					
**Female**	75 (40.8%)	27 (33.3%)	50 (39.1%)	52 (38.0%)	102 (38.5%)
**Male**	109 (59.2%)	54 (66.7%)	78 (60.9%)	85 (62.0%)	163 (61.5%)
**Age (median (IQR))**	65 (53–77)	75 (65–84)	65 (51–79)	70 (59–80)	68 (56–79)
**Comorbidities**					
**Alcohol abuse**	9 (4.9%)	0 (0.0%)	8 (6.2%)	1 (0.7%)	9 (3.4%)
**CAD**	49 (26.6%)	43 (53.1%)	31 (24.2%)	61 (44.5%)	92 (34.7%)
**Chronic lung disease**	31 (16.8%)	25 (31.2%)	22 (17.2%)	34 (25.0%)	56 (21.2%)
**Bronchiectasis**	10 (5.4%)	26 (32.1%)	5 (3.9%)	31 (22.6%)	36 (13.6%)
**Emphysema and fibrosis**	16 (8.8%)	25 (31.2%)	10 (7.9%)	31 (23.0%)	41 (15.7%)
**Diabetes mellitus**	42 (22.8%)	25 (30.9%)	26 (20.3%)	41 (29.9%)	67 (25.3%)
**Hypertension**	106 (57.9%)	50 (61.7%)	65 (51.2%)	91 (66.4%)	156 (59.1%)
**Obesity**	45 (24.9%)	12 (15.0%)	27 (21.3%)	30 (22.4%)	57 (21.8%)
**Smoker**	33 (17.9%)	12 (15.0%)	25 (19.5%)	20 (14.7%)	45 (17.0%)
**Symptoms**					
**Abdominal symptoms ^1^**	42 (22.8%)	8 (9.9%)	27 (21.1%)	23 (16.8%)	50 (18.9%)
**Cardiac symptoms ^2^**	29 (15.8%)	6 (7.4%)	21 (16.4%)	14 (10.2%)	35 (13.2%)
**Cough**	99 (53.8%)	37 (45.7%)	67 (52.3%)	69 (50.4%)	136 (51.3%)
**Dyspnea**	128 (69.6%)	68 (84.0%)	76 (59.4%)	120 (87.6%)	196 (74.0%)
**Fever**	122 (66.7%)	49 (60.5%)	84 (66.1%)	87 (63.5%)	171 (64.8%)
**Limb pain**	33 (17.9%)	5 (6.2%)	24 (18.8%)	14 (10.2%)	38 (14.3%)
**Weakness**	117 (63.6%)	54 (66.7%)	83 (64.8%)	88 (64.2%)	171 (64.5%)
**Ventilation**					
**None**	56 (30.8%)	0 (0.0%)	54 (42.5%)	2 (1.5%)	56 (21.4%)
**Nasal Oxygen**	87 (47.8%)	25 (31.2%)	67 (52.8%)	45 (33.3%)	112 (42.7%)
**Noninvasive**	39 (21.4%)	10 (12.5%)	6 (4.7%)	43 (31.9%)	49 (18.7%)
**Invasive**	0 (0.0%)	34 (42.5%)	0 (0.0%)	34 (25.2%)	34 (13.0%)
**ECMO**	0 (0.0%)	11 (13.8%)	0 (0.0%)	11 (8.1%)	11 (4.2%)

^1^ Abdominal symptoms: abdominal pain, nausea, vomiting, diarrhea, etc., ^2^ Cardiac symptoms: palpitations, blood pressure fluctuations, chest pain.

**Table 2 diagnostics-11-01616-t002:** Synopsis of subjective CT scan readings. The predominant findings in the CT scans were ground-glass opacities (89.1% of all patients) and consolidations (74.3% of all patients); pleural effusions and lymphadenopathy were rarely found. Patients who later needed invasive ventilation or ICU admission were more severely affected.

	No Need for Invasive Ventilation (*n* = 184)	Need for Invasive Ventilation (*n* = 81)	No Need for ICU Treatment (*n* = 128)	Need for ICU Treatment (*n* = 137)	Total (*n* = 265)
Subjective Classification of Pulmonary Involvement					
**None**	22 (12.0%)	2 (2.5%)	18 (14.1%)	6 (4.4%)	24 (9.1%)
**Minor**	72 (39.1%)	20 (24.7%)	58 (45.3%)	34 (24.8%)	92 (34.7%)
**Moderate**	74 (40.2%)	32 (39.5%)	47 (36.7%)	59 (43.1%)	106 (40.0%)
**Major**	16 (8.7%)	27 (33.3%)	5 (3.9%)	38 (27.7%)	43 (16.2%)
**Consolidations**					
**None**	61 (33.2%)	7 (8.6%)	46 (35.9%)	22 (16.1%)	68 (25.7%)
**Minor ^1^**	85 (46.2%)	38 (46.9%)	64 (50.0%)	59 (43.1%)	123 (46.4%)
**Moderate ^1^**	28 (15.2%)	25 (30.9%)	15 (11.7%)	38 (27.7%)	53 (20.0%)
**Major ^1^**	10 (5.4%)	11 (13.6%)	3 (2.3%)	18 (13.1%)	21 (7.9%)
**Ground-Glass Opacities (GGOs)**					
**No**	24 (13.0%)	5 (6.2%)	17 (13.3.%)	12 (8.8%)	29 (10.9%)
**Minor ^1^**	65 (35.3%)	19 (23.5%)	47 (36.7%)	37 (27.0%)	84 (31.7%)
**Moderate ^1^**	72 (39.1%)	29 (35.8%)	50 (39,1%)	52 (37.2%)	101 (38.1%)
**Major ^1^**	23 (12.5%)	28 (34.6%)	14 (10.9%)	37 (27.0%)	51 (19.2%)
**Effusions**					
**No**	165 (89.7%)	59 (72.8%)	117 (91.4%)	107 (78.1%)	224 (84.8%)
**Minor**	15 (8.2%)	19 (23.5%)	10 (7.8%)	24 (17.5%)	34 (12.8%)
**Moderate**	3 (1.6%)	3 (3.7%)	1 (0.8%)	5 (3.6%)	6 (2.3%)
**Major**	1 (0.5%)	0 (0%)	0 (0%)	1 (0.7%)	1 (0.4%)
**Lymphadenopathy**	35 (19.0%)	21 (25.9%)	21 (16.4%)	35 (25.5%)	56 (21.1%)
**Distribution**					
**Central**	3 (1.6%)	3 (3.7%)	3 (2.3%)	3 (2.2%)	6 (2.3%)
**Diffuse**	13 (7.1%)	27 (33.3%)	5 (3.9%)	35 (25.5%)	40 (15.1%)
**No pattern**	21 (11.4%)	2 (2.5%)	17 (13.3%)	6 (4.4.%)	23 (8.7%)
**Peribronchial**	72 (39.1%)	21 (25.9%)	52 (40.6%)	41 (29.9%)	93 (35.1%)
**Peripheral**	75 (40.8%)	28 (34.6%)	51 (39.8%)	52 (38.9%)	103 (38.9%)
**Predominant Localization**					
**Apical**	18 (9.7%)	6 (7.4%)	11 (8.6%)	13 (9.5%)	23 (8.7%)
**Basal**	82 (44.6%)	33 (40.7%)	59 (46.1%)	56 (40.9%)	115 (43.4%)
**Medial**	(9 (4.9%)	6 (7.4%)	7 (5.5%)	8 (5.8%)	15 (5.7%)
**No pattern**	21 (11.4%)	5 (6.2%)	17 (13.3%)	9 (6.6%)	26 (9.8%)
**Posterolateral**	54 (29.3%)	31 (38.3%)	34 (26.6%)	51 (37.2%)	85 (32.1%)

^1^ Classification of affected lung area: minor = 0–33%, moderate = 34–66%, major = 67–100%.

**Table 3 diagnostics-11-01616-t003:** Simple semiquantitative (SSQ) assessment. Total affected lung area was calculated from all three levels at which involvement was measured. Patients with need for ICU admission or invasive ventilation were affected more severely, especially the cranial segments of the lung.

	No Need for Invasive Ventilation (*n* = 184)	Need for Invasive Ventilation (*n* = 81)	No Need for ICU Treatment (*n* = 128)	Need for ICU Treatment (*n* = 137)	Total (*n* = 265)
**Aortic arch (% of lung area affected)**	5.4 (0.0–19.7)	20.3 (7.9–42.8)	2.7 (0.0–11.5)	19.6 (5.8–37.6)	9.9 (0.7–25.2)
**Tracheal bifurcation (% of lung area affected))**	10.6 (1.7–21.3)	24.0 (9.8–51.3)	6.2 (0.5–16.0)	20.1 (9.7–44.1)	12.8 (3.7–31.6)
**Inferior end of xiphoid (% of lung area affected))**	14.6 (4.8–27.8)	32.8 (14.5–57.9)	12.0 (4.4–24.1)	24.7 (13.2–49.5)	18.8 (7.3–37.5)
**Total affected lung area (%)**	11.7 (4.7–24.4)	25.5 (14.5–52.3)	8.2 (3.6–19.6)	22.6 (12.0–43.3)	15.3 (6.5–31.1)

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
