# Peer review of "CT-Based Risk Stratification for Intensive Care Need and Survival in COVID-19 Patients—A Simple Solution"

_diagnostics, 2021, doi:10.3390/diagnostics11091616_

Round 1

Reviewer 1 Report

I read with great interest the paper. I find it well wrote. Below my minor suggestions

  1. Introduction: updata data on SARS CoV2 global burden
  2. Methods and results: clear and with high quality presentation
  3. Discussion: discuss better the role of age and comorbidity on outcome (see and cite COVID-19 RISK and Treatments (CORIST) Collaboration. Lopinavir/Ritonavir and Darunavir/Cobicistat in Hospitalized COVID-19 Patients: Findings From the Multicenter Italian CORIST Study ) and the role of heparin (Heparin in COVID-19 Patients Is Associated with Reduced In-Hospital Mortality: The Multicenter Italian CORIST Study. Thromb Haemost. 2021 Aug;121(8):1054-1065.)
  4. Furthermore, discuss also the possible role of ultrasound in lung diseases as pneumonia and it is possibile role in COVID 19 monitoring
  5. Conclusion: give same public health proposal that come from your very interesting paper

Reviewer 2 Report

Hosse et al. submit a paper in which they analyse the relationship between different approaches (subjective and simple semi-quantitative method) and the risk of ICU admission, invasive ventilation and death for COVID-19-affected patients.

The methodology and the analysis are classical but well-presented and clean. The article is well written. Although there are some comments to be made.

  • Lines 107-112: the authors declare that to avoid consequences coming from existing living wills of their cohort, they chose medical indication to determine the endpoint. Therefore, no information are given about their inclusion or exclusion in cases of death. How the authors avoid this bias?
  • Subjective assessment is too variable to be used as an evaluation method within a scientific context. Even more so that it was not explained by whom the images were interpreted (radiologist or clinician? Chest expert or generic?). To avoid this bias, blinded and after randomization, an interpretation could be carried out by several operators with subsequent estimation of the inter-operator variability index, or, considering what was stated on lines 341-342, two examinations by the same operator.

This limit should be further emphasized considering that it was used as a comparison to underline the importance and effectiveness of the SSQ method.

  • Diagnostic performance: is the choice of the three sections in which to carry out the measurements by SSQ arbitrary or does it have its own rationale? It is not clear.

Round 2

Reviewer 2 Report

Dear Autor, 

I greatly appreciated the explanations given, however they have not been made explicit in the text (with the exception of the extension of the paragraph "Limitations"), therefore, the same perplexities would be detectable by any reader.